# Environmental Selection and Biogeography Shape the Microbiome of Subsurface Petroleum Reservoirs

Daniel A. Gittins,[a] Srijak Bhatnagar,[a,b] Casey R. J. Hubert[a]

aDepartment of Biological Sciences, University of Calgary, Calgary, Canada
bFaculty of Science and Technology, Athabasca University, Athabasca, Canada

**ABSTRACT** Petroleum reservoirs within the deep biosphere are extreme environments inhabited by diverse microbial communities and represent biogeochemical hot spots in the subsurface. Despite the ecological and industrial importance of oil reservoir microbiomes, systematic study of core microbial taxa and their associated genomic attributes spanning different environmental conditions is limited. Here, we compile and compare 343 16S rRNA gene amplicon libraries and 25 shotgun metagenomic libraries from oil reservoirs in different parts of the world to test for the presence of core taxa and functions. These oil reservoir libraries do not share any core taxa at the species, genus, family, or order levels, and *Gammaproteobacteria* was the only taxonomic class detected in all samples. Instead, taxonomic composition varies among reservoirs with different physicochemical characteristics and with geographic distance highlighting environmental selection and biogeography in these deep biosphere habitats. Gene-centric metagenomic analysis reveals a functional core of metabolic pathways including carbon acquisition and energy-yielding strategies consistent with biogeochemical cycling in other subsurface environments. Genes for anaerobic hydrocarbon degradation are observed in a subset of the samples and are therefore not considered to represent core functions in oil reservoirs despite hydrocarbons representing an abundant source of carbon in these deep biosphere settings. Overall, this work reveals common and divergent features of oil reservoir microbiomes that are shaped by and responsive to environmental factors, highlighting controls on subsurface microbial community assembly.

**IMPORTANCE** This comprehensive analysis showcases how environmental selection and geographic distance influence the microbiome of subsurface petroleum reservoirs. We reveal substantial differences in the taxonomy of the inhabiting microbes but shared metabolic function between reservoirs with different *in situ* temperatures and between reservoirs separated by large distances. The study helps understand and advance the field of deep biosphere science by providing an ecological framework and footing for geologists, chemists, and microbiologists studying these habitats to elucidate major controls on deep biosphere microbial ecology.

**KEYWORDS** environmental microbiology, geomicrobiology, metagenomics, microbial ecology

The subsurface biosphere is the largest microbial habitat on Earth (1). Microbes inhabiting this environment play an important role in mediating global scale biogeochemical cycling of elements and nutrients (2). Understanding the factors that shape microbial community structure and functional potential helps to uncover these processes and understand how geology and biology interact. The deep biosphere includes diverse terrestrial and marine habitats such as aquifer systems, basaltic ocean crust, buried sediments, and petroleum reservoirs. The latter represents an interesting microbial setting in the subsurface context in that energy-rich petroleum compounds

Address correspondence to Daniel A. Gittins, daniel.gittins@ucalgary.ca, or Casey R. J. Hubert, chubert@ucalgary.ca.

The authors declare no conflict of interest.

offer substrates and electron donors, while at the same time hydrocarbons can be toxic to microorganisms (3, 4). Oil reservoirs are generally considered extreme environments as a result of high temperature, high pressure, and millions of years of isolation from the Earth's surface (5). Despite this, petroleum reservoirs contain an order of magnitude more microorganisms than surrounding sediments at similar depths (6), indicating that these systems may be deep biosphere "hot spots" of microbial activity.

The physiology and metabolism of the petroleum reservoir microbiome influence the physicochemical properties of these environments. Crude oil biodegradation changes the composition and physical properties of both liquid and gaseous components of petroleum via the metabolism of hydrocarbons and other compounds (7). Biodegradation of petroleum hydrocarbons under methanogenic conditions over geological timescales is catalyzed *in situ* by consortia of bacteria and archaea, altering both the chemistry of the remaining oil as well as the levels of $CO_2$ and $CH_4$ (8, 9). Interestingly, this oil-altering biogeochemistry appears to only take place in reservoirs that have a burial history that has not included depths hotter than 80 to 90℃ that inactivate these populations by heat sterilization (10). In situations where sulfate is present, anaerobic respiration of sulfate to sulfide, known as reservoir souring, can be coupled to the oxidation of hydrocarbons directly, or indirectly via oxidation of organic acids or hydrogen derived from crude oil biodegradation (11–13). Accurate characterization of these metabolic processes catalyzed by the petroleum reservoir microbiome enables useful predictions of reservoir conditions.

Microbial communities in nature are typically highly complex, comprising thousands of species. A "core microbiome" is considered to be the consistent members of complex microbial assemblages present in a given habitat type, as confirmed by observations of common taxa across several different sampling sites (14–16). Identifying the occurrence of a core microbiome is important for understanding ecosystem health, functioning, and changes. Microorganisms occurring at a higher frequency are likely important for normal biogeochemical processes, such that the core microbiome can define dominant metabolic processes and microbial interactions (17–19). Temporal and spatial heterogeneity in microbiota can also occur within sample sets of a common habitat type, pointing to dynamic variation in functioning across complex microbiomes (20). In petroleum reservoirs, identifying shared and nonshared taxa can increase understanding of the ecosystem and may also guide microbiology-inspired engineering innovations to manipulate these subsurface communities and the biogeochemical processes they mediate.

The most common method for microbial community profiling is to sequence the 16S rRNA gene enabling taxonomic identification of community members. Shotgun metagenome sequencing is also being increasingly applied to many ecosystems, adding functional profiles and advancing understanding of the microbial ecology of a given environment. To obtain a comprehensive view of oil reservoirs as an ecosystem, we provide a meta-analysis of 16S rRNA genes (amplicon sequencing) and metabolic genes (metagenomic sequencing) from a compiled data set of published microbial assessments of hydrocarbon reservoir fluids from a globally diverse set of petroleum reservoirs. We test the hypothesis that petroleum reservoirs contain a core microbiome with respect to both taxonomy and biogeochemical functions. The results reveal insights into the shared functions present across the disparate petroleum reservoirs, highlighting the role of environmental selection in these subsurface settings.

## RESULTS

**Overview of the petroleum reservoir microbiome.** A total of 2,473 phylotypes were inferred from high-throughput 16S rRNA gene amplicon sequencing (Table S5 in the supplemental material) of 201 subsurface oil reservoir samples. Phylotypes encompassed 96 phyla comprising 2,283 genera in total. At the phylum level, *Proteobacteria* (30%), *Euryarchaeota* (19%), *Halobacterota* (15%), *Firmicutes* (9%), and *Campylobacteria* (5%) were most abundant. Classes and genera present at >1% average relative sequence abundance all belong to the 10 most abundant phyla in the data set (Fig. 1). At the genus level, *Thermococcus*, *Methanosaeta*, and *Methanothermobacter* cumulatively

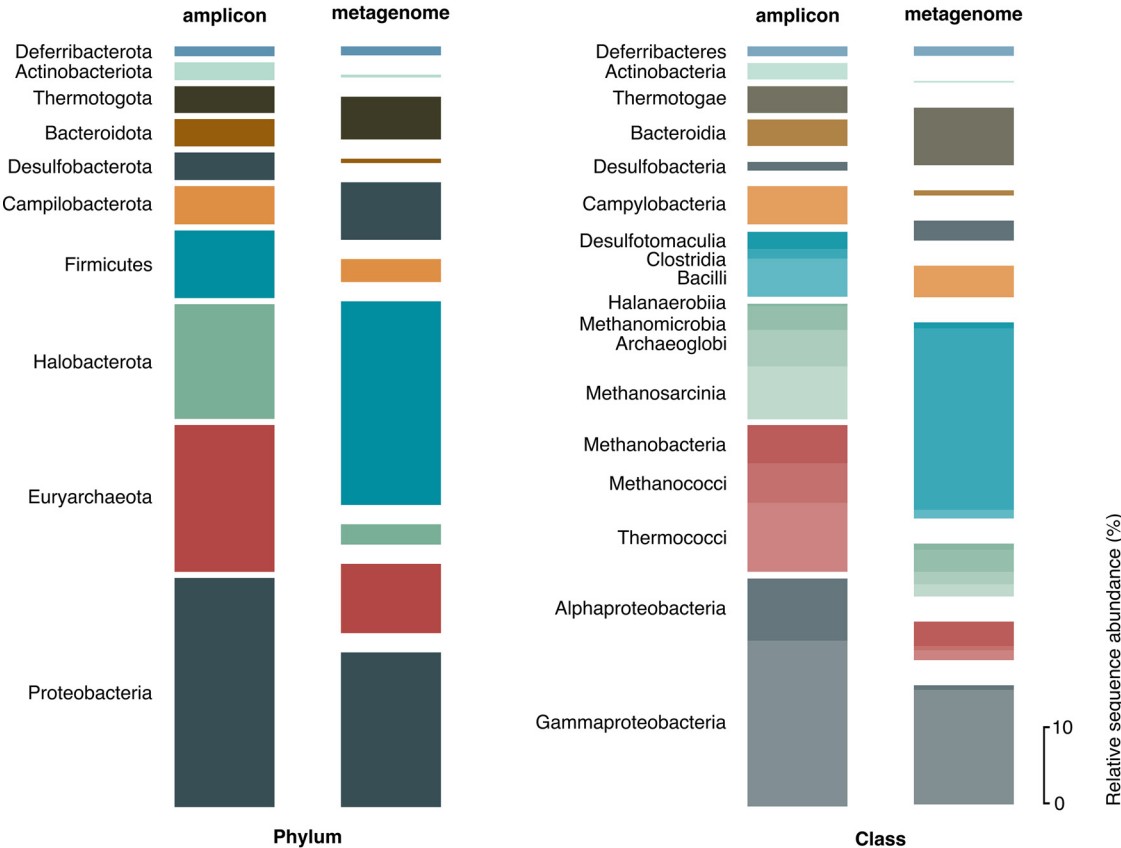

**FIG 1** Petroleum reservoir microbiome. Average relative sequence abundance of the 10 most abundant phyla and all classes >1% relative sequence abundance in 295 high-throughput 16S rRNA gene amplicon sequencing libraries. The corresponding abundance of the same phyla and classes in 25 metagenome libraries is also shown. Phyla represent 93% and 78% of the 16S and metagenomes libraries, respectively. Classes represent 90% and 70%, respectively.

account for 4 to 8% of the sequences. Oil reservoir clone libraries (48 libraries comprising 2,850 reads) support findings from the more extensive high-throughput 16S rRNA gene data, with *Halobacterota*, *Proteobacteria*, and *Firmicutes* accounting for 26, 20 and 13%, respectively, at the phylum level. Oil reservoir metagenomes corroborate 16S rRNA gene-sequencing approaches in terms of the most prevalent taxonomic groups but highlight differences in relative abundance estimates (Fig. 1). The most abundant phyla in metagenome libraries were *Firmicutes* (27%), *Proteobacteria* (20%), *Euryarchaeota* (9%), *Desulfobacterota* (8%), and *Thermotogota* (6%), corresponding with the most abundant genera in the metagenomes being *Caminicella*, *Pseudomonas*, *Desulfonauticus*, *Petrotoga*, and *Thermoanaerobacter*, each accounting for 4 to 15% of the genus-level assignments. Inconsistencies in community composition using these different DNA sequencing strategies may reflect true variations between the reservoirs sampled (few reservoirs in this study had both amplicon and metagenome sequencing applied) or could reflect well-known methodological differences such as preferential amplification of certain phylotypes during 16S rRNA gene sequence library preparation skewing the reported community composition.

Assessing whether or not petroleum reservoirs around the world harbor a core microbiome revealed that the bacterial phylum *Proteobacteria* was observed across all 201 samples assessed by high-throughput 16S rRNA gene amplicon sequencing (Table S5), while *Firmicutes*, *Bacteroidota*, *Halobacterota*, *Actinobacteriota*, and *Desulfobacterota* were observed across >90% of the samples. At a finer taxonomic resolution, *Gammaproteobacteria* were detected in all samples, but no other taxonomic class and no taxonomic orders were consistently observed. The absence of a core microbiome at the order level was substantiated in the analysis of libraries prepared with primers for the amplification of archaeal 16S rRNA

genes, which showed only *Halobacterota* and *Euryarchaeota* are observed in all libraries. Metagenomes revealed that at the phylum level *Firmicutes* and *Proteobacteria*, along with *Euryarchaeota*, were observed in all samples, and at the genus level *Thermovirga*, *Halanaerobium*, and *Petrotoga* were observed in more than 75% of the samples. Considering oil reservoirs produced with primary recovery, the genus *Pseudomonas* was observed across all samples ($n = 91$) analyzed using high-throughput 16S rRNA gene amplicon sequencing; this Gammaproteobacterial genus likely represents the best candidate for a core microbial population in undisturbed reservoir ecosystems. Comparative analysis of reservoirs produced by secondary recovery methods like water injection showed *Gammaproteobacteria* were detected across all samples, whereas no finer-resolution taxa were consistently observed. Based on defining core taxa at the species level (16), the sequencing results compiled here falsify the hypothesis that oil reservoirs harbor a core microbiome based on taxonomy.

**Environmental influences on oil reservoir microbial community structure.** Depth and, by association, temperature of petroleum reservoir samples spanning six continents ranged from 270 to 3,550 m below surface (mbsf; defined as the seabed in offshore settings) and 8 to 110°C (Table S1). Alpha diversity and depth are negatively correlated in high-throughput amplicon libraries (Spearman's rank correlation $r = -0.47$ and $-0.49$, respectively, $P < 0.05$), with the deepest and hottest samples ($>2,000$ m and 74°C average reservoir temperature; $n = 38$ samples) having a Shannon diversity index $H'$ of 1.6 on average, compared to an average $H'$ of 2.8 in the shallowest and coolest reservoirs ($<500$ m and 11°C average reservoir temperature; $n = 75$ samples) (Table S1; Table S3).

Reservoir temperature and depth, as with alpha diversity, are correlated with greater variation in high-throughput 16S rRNA gene-based community composition, such that as the difference in temperature and depth of two compared samples increases, community composition variance increases (Fig. 2; Table S4). Assessments of phylotype presence and abundance for distinct reservoir temperature intervals using *indicspecies* tests (reference 21; Table S6) showed that most members within the commonly observed phyla (e.g., *Proteobacteria*, *Firmicutes*, *Bacteroidota*, *Halobacterota*, *Actinobacteriota*, and *Desulfobacterota*) were not significantly associated with an individual temperature interval (i.e., 10°C intervals spanning 0 to 90°C). In contrast, at the genus level, *Thermococcus* and *Petrotoga* demonstrated strong correlations with temperature, being prevalent community members in reservoirs between 71 to 80°C and 81 to 90°C, respectively. This was consistent with results from metagenomes, particularly for *Petrotoga* (Table S5).

Oil reservoirs undergoing secondary recovery operations (197 amplicon libraries from 110 different samples) are less diverse (average $H' = 1.7$) than reservoirs sampled during primary recovery (91 samples, 98 libraries; $H' = 2.8$) (Table S1; Table S3). Whereas primary oil recovery is mainly driven by *in situ* reservoir pressure, secondary oil recovery involves reestablishing pressure in the reservoir after the initial primary production phase by injecting fluids (e.g., seawater). Primary or secondary oil recovery methods accounted for only a small fraction of the variation in microbial community composition (Table S4). However, significant correlations were apparent between geographic location and reservoir microbiome similarity (Fig. 2B). As such, reservoirs in similar locations tend to have more similar microbial community compositions than reservoirs separated by large geographic distances. This distance decay relationship is more prominent if only reservoirs produced by primary recovery are considered (Mantel, archaeal $r = 0.87$; universal $r = 0.80$; $P < 0.05$) compared to when reservoirs produced by secondary recovery are incorporated into geographic distance analysis (Mantel, archaeal $r = 0.44$; universal $r = 0.30$; $P < 0.05$). This suggests a biogeography especially in pristine oil reservoir deep biosphere environments that is diminished after populations shift in response to secondary recovery.

**Functional potential of the oil reservoir microbiome.** Metagenomes from 25 samples associated with both primary and secondary recovery and from reservoirs ranging from 457 to 3,350 mbsf and 26 to 102°C were compiled for community-wide gene prediction to assess the functional potential of the oil reservoir microbiome. Analysis of 6,992 unique Kyoto Encyclopedia of Genes and Genomes (KEGG) orthology (KO) assignments shows the functional composition of communities is conserved across

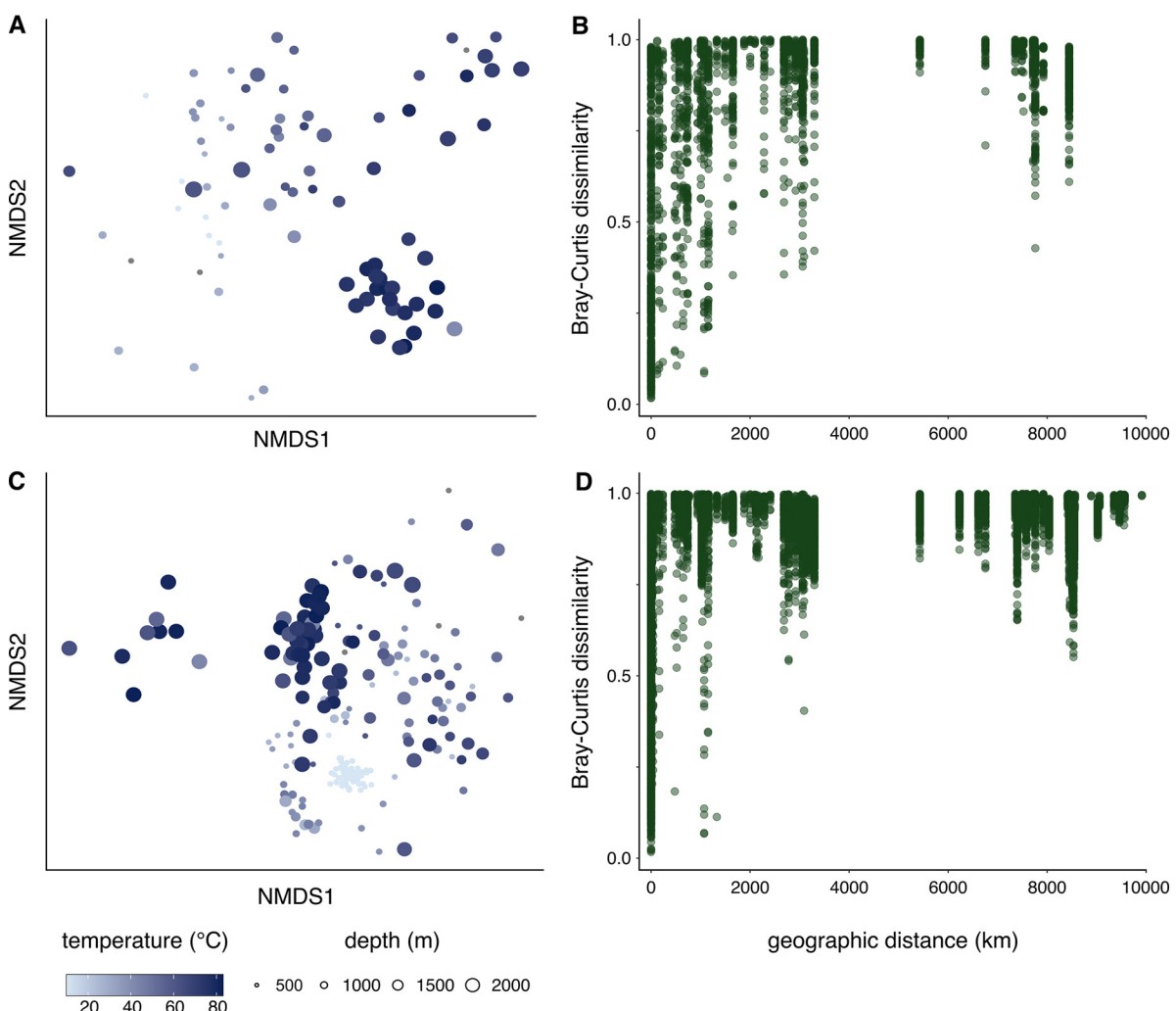

**FIG 2** Relationship between microbial community composition and environmental factors. High-throughput 16S rRNA gene libraries were grouped based on the use of archaea-specific primers (A and B) or universal primers (C and D) to perform Bray-Curtis community dissimilarity comparisons based on temperature, depth, and geographic distance. A geographic distance of "0" indicates reservoirs at the same location. Libraries prepared using archaeal primers (A) exhibit greater dissimilarity between samples as a function of temperature (Mantel, $r = 0.55$) and depth (Mantel, $r = 0.55$) than libraries prepared with universal primers (C) (Mantel, $r = 0.33$ and 0.43, respectively), although both represent significant dissimilarity ($P < 0.05$). Visual trends as well as statistical testing (Mantel, $P < 0.05$) in amplicon libraries using archaeal (B) and universal (D) primers indicate that geographically distant reservoirs have more dissimilar microbial communities.

samples from different studies (Fig. 3A; Table S4). Unlike 16S rRNA taxonomy-based associations of microbiomes with environmental factors (Fig. 2 and 3B), no significant correlation was observed between gene composition and reservoir type (i.e., primary versus secondary recovery), temperature, depth, or geographic location (Table S4). This is consistent with a functional core set of genes in subsurface oil reservoirs.

Reservoir metagenomes were compared to better understand functional potential for carbon acquisition and energy conservation (Fig. 4; Table S7). Diverse capabilities for carbohydrate, peptide, and lipid catabolism, as well as carbon fixation, are widespread in oil reservoirs. Mixed-acid fermentation appears to be a universal strategy. Acetate production from pyruvate is observed in all metagenomes, consistent with acetate often being measured in oil field waters (22, 23) and acetogenesis being important in the deep biosphere in general (24). Energy conservation through anaerobic respiration is another widespread metabolism. Genes required for the reduction of sulfate to sulfide (*sat*, *aprAB*, and *dsrAB*) were observed in 24 of the 25 metagenomes. Reductases required for nitrate reduction to nitrite, dissimilatory nitrate reduction to

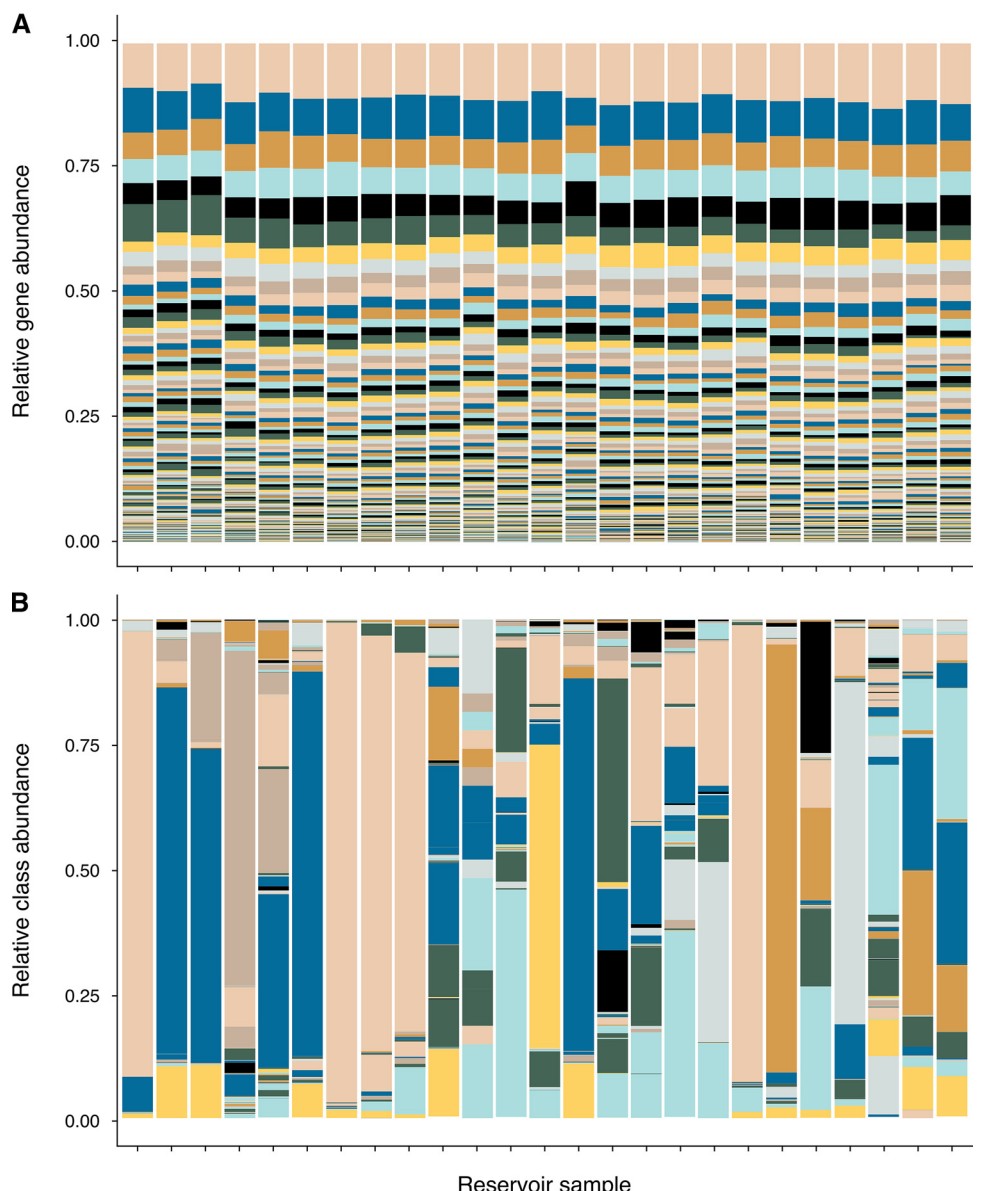

**FIG 3** Metabolic gene and taxonomic composition of samples from different reservoirs. (A) Relative abundance of genes (represented by unique KEGG orthologs and grouped by KEGG Level C pathway) in metagenome sequence libraries from 25 different petroleum reservoir samples. Ortholog counts were subsampled to the lowest total number of KEGG-annotated sequences ($n = 7,474$) across all samples in the data set. (B) Relative abundance of taxonomic groups (class level) in the same 25 metagenome sequence libraries. The similar gene composition profiles and variable taxonomy-based associations reflect a shared genetic potential of the petroleum reservoir microbiome regardless of prevailing taxa in these habitats (Table S4).

ammonia (DNRA), or denitrification to $N_2$ were detected in 18, 15, and 8 of the metagenomes, respectively. Observations of sulfate and nitrate metabolism genes do not correlate with primary or secondary recovery practices, indicating that these respiratory pathways are universal features of indigenous microbial communities. Sulfide: quinone oxidoreductase (*sqr*), which catalyzes the oxidation of sulfide to elemental sulfur, was detected in all samples. Sulfide oxidation can be coupled to DNRA or denitrification, which these oil reservoir microbiomes generally exhibit capability for, as noted above.

Hydrogen metabolism in oil reservoirs can be linked to both fermentative and respiratory pathways (Fig. 4; Table S7). Membrane-bound group 1b respiratory $H_2$-uptake

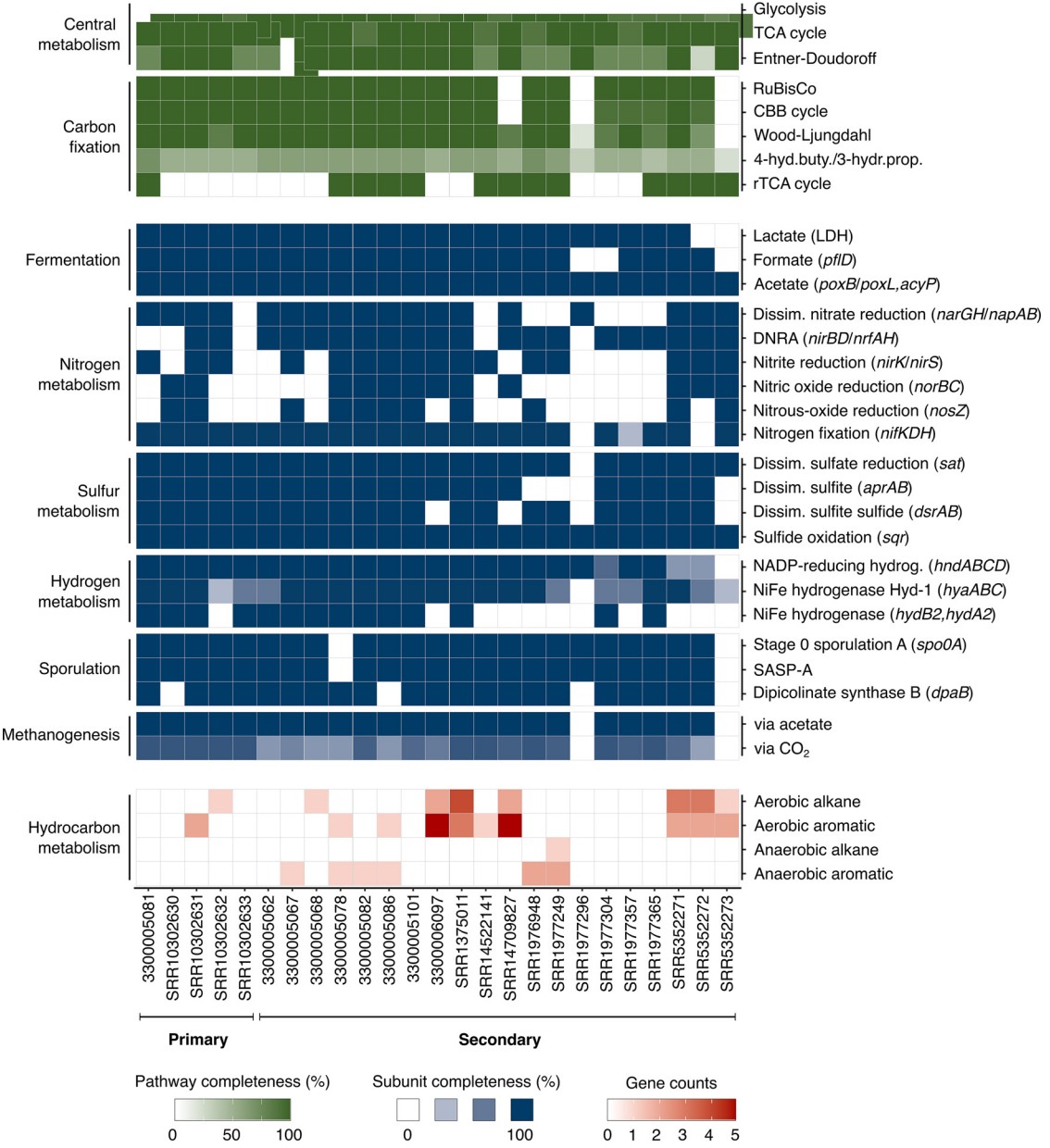

**FIG 4** Metabolic potential of the petroleum reservoir microbiome. Occurrence of metabolic genes and pathways in 25 metagenomes from nine reservoirs. Complete gene lists can be found in Table S7.

hydrogenases (*hydB2* and *hydA2*) and group 1d oxygen-tolerant NiFe hydrogenase (*hyaABC*) are both observed in the metagenomes. These enzymes pair hydrogen oxidation with the liberation of electrons for anaerobic respiration (25). Consistent cooccurrence of *hyaABC* with the near universal dissimilatory sulfate reduction gene sulfate adenylyltransferase (*sat*) suggests potential coupling of this form of hydrogen oxidation with sulfate respiration. Genes encoding the group 3b cytosolic NiFe hydrogenase (*hndABCD*) were present in nearly all metagenomes. This enzyme can work bidirectionally but generally couples reoxidation of NAD(P)H to $H_2$ generation via fermentation (25). The diversity of hydrogenases in oil reservoir metagenomes suggests that both hydrogen production and consumption may be widespread features of oil reservoir biogeochemistry.

Microbial dormancy in the form of endospore formation is considered to be an important process in the deep biosphere (26). Key marker genes for sporulation are

widespread in these subsurface oil reservoir metagenomes (Fig. 4; Table S7). These include the *spo0A* master transcriptional response regulator that controls entry into sporulation and genes encoding small acid-soluble protein (SASP-A) and dipicolinic acid (*dpaB*) synthesis, which are important in protecting spore DNA against damage. This finding is consistent with the *Firmicutes* phylum (to which known endospore-forming microbes belong) demonstrating the highest average relative abundance in oil reservoir 16S rRNA gene and metagenome libraries (Fig. 1).

**Hydrocarbon biodegradation potential.** To identify the potential for microbial degradation of crude oil, functional marker genes encoding enzymes that initiate aerobic or anaerobic hydrocarbon biodegradation, by activating either alkane or aromatic compounds, were examined using a set of 37 hidden Markov models (reference 27; Fig. 4; Table S7). Of the 28 aerobic and 9 anaerobic marker genes tested, 17 were identified in 16 of the 25 metagenomes. Among these 17 genes, 14 are associated with the aerobic biodegradation of hydrocarbons. Genetic potential for the aerobic oxidation of *n*-alkanes with chain lengths $C_{10}$ to $C_{16}$ by *alkB* alkane hydroxylase and short- and medium-chain-length *n*-alkanes by the cytochrome P450 CYP153 family alkane hydroxylase was prevalent. Genes involved in the aerobic degradation of long-chain alkanes (e.g., *almA* and *ladA/B*) and polyaromatic hydrocarbons (e.g., *ndoB/C* and *dszC*) were less frequently observed.

Anaerobic degradation of *n*-alkanes and aromatic hydrocarbons can be initiated by their addition to fumarate to form succinates. Sequences encoding the catalytic subunit of benzylsuccinate synthase (*bssA*) were detected in six samples from reservoirs in the North Sea and North Slope Alaska, with the latter also hosting the catalytic subunits of alkylsuccinate synthase (*assA*) and naphthylmethylsuccinate synthase (*nmsA*). Metabolisms encoded by these genes can couple hydrocarbon degradation to the reduction of an electron acceptor such as sulfate or to fermentative or syntrophic metabolisms that proceed in conjunction with methanogenesis. Genes involved in acetoclastic and hydrogenotrophic methanogenesis that facilitate this were detected in 23 of the 25 metagenomes. Similarly, the gene encoding the group 3a $F_{420}$ hydrogenase (*frhB*; KEGG ortholog KO00441), which directly couples oxidation of hydrogen to the reduction of $F_{420}$ during methanogenesis (28), is widespread, being detected in 23 of the 25 metagenomes (Table S7).

## DISCUSSION

**Petroleum reservoirs lack a core microbiome.** By compiling 295 amplicon libraries from 41 oil reservoirs totaling 53 million sequence reads, this study falsified the hypothesis that oil reservoirs contain a core microbiome, as defined by the presence of common species in all samples (15, 16, 18). Taxonomic assessments showed that even at the order level not all of the samples included representatives of the most prevalent groups, such as *Burkholderiales*, *Rhizobiales*, and *Pseudomonadales*. Taxonomic classification of ribosomal rRNA genes from metagenomic libraries in a smaller number of samples supported observations from amplicon libraries and indicated that no species are universally present in oil reservoirs.

Instead of a core microbiome in oil reservoirs, environmental factors result in niche habitat partitioning whereby taxonomically different organisms comprise the microbiome in oil reservoir habitats featuring different physicochemical conditions. Metagenomic sequencing libraries from 25 reservoir samples reveal that different oil reservoirs have shared biogeochemical functions (Fig. 3A; Table S4). Redundant functional potential in different oil reservoirs demonstrates trait selection in these settings despite the absence of a taxonomic core (Fig. 3B). Diverse genes encode functions for processes such as fermentation, sulfate reduction, hydrocarbon biodegradation, and methanogenesis in the oil reservoir microbiome. These and other metabolic functions were highlighted in a similar meta-analysis of oil reservoir metagenome-assembled genomes (MAGs) by Hidalgo et al. (29) that focused on nine different oil reservoirs in China, Alaska's North Slope, and offshore Brazil. That genome-centric approach revealed both core and environment-specific functions but also included observations that differ from those in the gene-centric analysis presented here with respect to the metabolisms noted above. Some inconsistency between

these two metagenome analysis strategies may be expected to arise based on preferential assembly of the most abundant organisms in genome-centric analyses resulting in an associated loss of unbinned sequence information in the process. For example, MAGs containing genes for dissimilatory sulfate reduction were identified in just over half of the oil fields in the genome-centric study (29) whereas in the analysis performed here, by avoiding genome binning, sulfate reduction genes were identified in 24 out of 25 samples. Similarly, genome-centric analysis suggested that sulfide oxidation was not widespread in oil fields (29), whereas the less restrictive analysis of unassembled contigs in the present study suggests a universal genetic potential for sulfide oxidation in oil reservoirs.

Genetic capacity for carbohydrate, peptide, and lipid metabolism, uncovered in the present study, indicates that the degradation of microbial necromass and sedimentary detrital material could be an important process in petroleum reservoirs or their precursor sediments. This is consistent with the role of residual organic matter recycling in the deep biosphere in general (30–32). The widespread occurrence of genes involved in acetogenesis in these metagenomes is also consistent with acetogens being important in anaerobic subsurface environments (24). Diversity and prevalence of both respiratory and fermentative pathways indicate these may be cooccurring or used sequentially by individual organisms in response to changing conditions. It is likely that fermentation products contribute substrates for further sulfate reduction or methanogenesis, depending on prevailing redox conditions. Not surprisingly, genes for sulfate reduction and methanogenesis were widespread in the oil reservoir metagenomes examined here. The potential for respiration may also explain the rapid transition to souring and souring control scenarios following the introduction of sulfate-rich seawater and injected nitrate, respectively.

**Is hydrocarbon biodegradation universal in petroleum reservoirs?** In pristine oil reservoirs, before secondary recovery or even reservoir discovery, the absence of oxygen or other electron acceptors dictates that methanogenic conditions should prevail in many settings. In this context, fermentation reactions that initiate anaerobic hydrocarbon biodegradation couple with methanogenic archaea rapidly consuming acetate, $CO_2$, and hydrogen to ensure thermodynamic feasibility of syntrophic partnerships (33, 34). Recent work demonstrates that similar metabolism (conversion of oil to methane) can also be facilitated by oil reservoir methanogens in the absence of a syntrophic partner (35). The genetic potential for acetoclastic and hydrogenotrophic methanogenesis was identified in all of the oil reservoirs examined here, consistent with the understanding that this may be a default biogeochemical regime in pristine petroleum-bearing sediments. This view is supported by thermodynamic modeling (34), carbon dioxide and methane stable carbon isotopic measurements (8), and radiotracer experiments (36) that predict hydrogenotrophic $CO_2$ reduction to be the primary route for crude oil hydrocarbon biodegradation in oil reservoirs.

The hydrocarbon activation genes assA, bssA, and nmsA were only detected in one-quarter of the oil reservoirs examined by metagenome sequencing in this study. A similar result was reported by Hidalgo et al. (29) who found only two bssA genes in 148 MAGs from different oil reservoirs using KEGG assignments. The discrepancy of widespread genomic potential for methanogenesis and geochemical evidence of biogenic methane production in reservoirs (37, 38) on the one hand, and limited occurrences of anaerobic hydrocarbon degradation genes on the other, hints that other mechanisms for anaerobic hydrocarbon biodegradation (39–41) or as-yet undetected hydrocarbon activation genes in anoxic petroleum reservoirs. Analysis of variants of glycyl-radical enzymes proposed to mediate anaerobic alkane biodegradation via addition to fumarate reveals a clade of genes encoding alkylsuccinate synthase (assA) that diverge from canonical assA found in *Proteobacteria* (Fig. 5). These assA-like pyruvate formate-lyase (pflD) genes are found in taxa from different petroleum reservoirs, including *Archaeoglobus fulgidus* (42, 43), [U]*Petromonas tenebris* (44), and *Thermococcus sibiricus* (45). In the reservoir metagenomes assessed here 16 out of the 25 libraries contained homologs of assA-like pyruvate formate-lyase (pflD) genes (Fig. 5), pointing to the underexplored

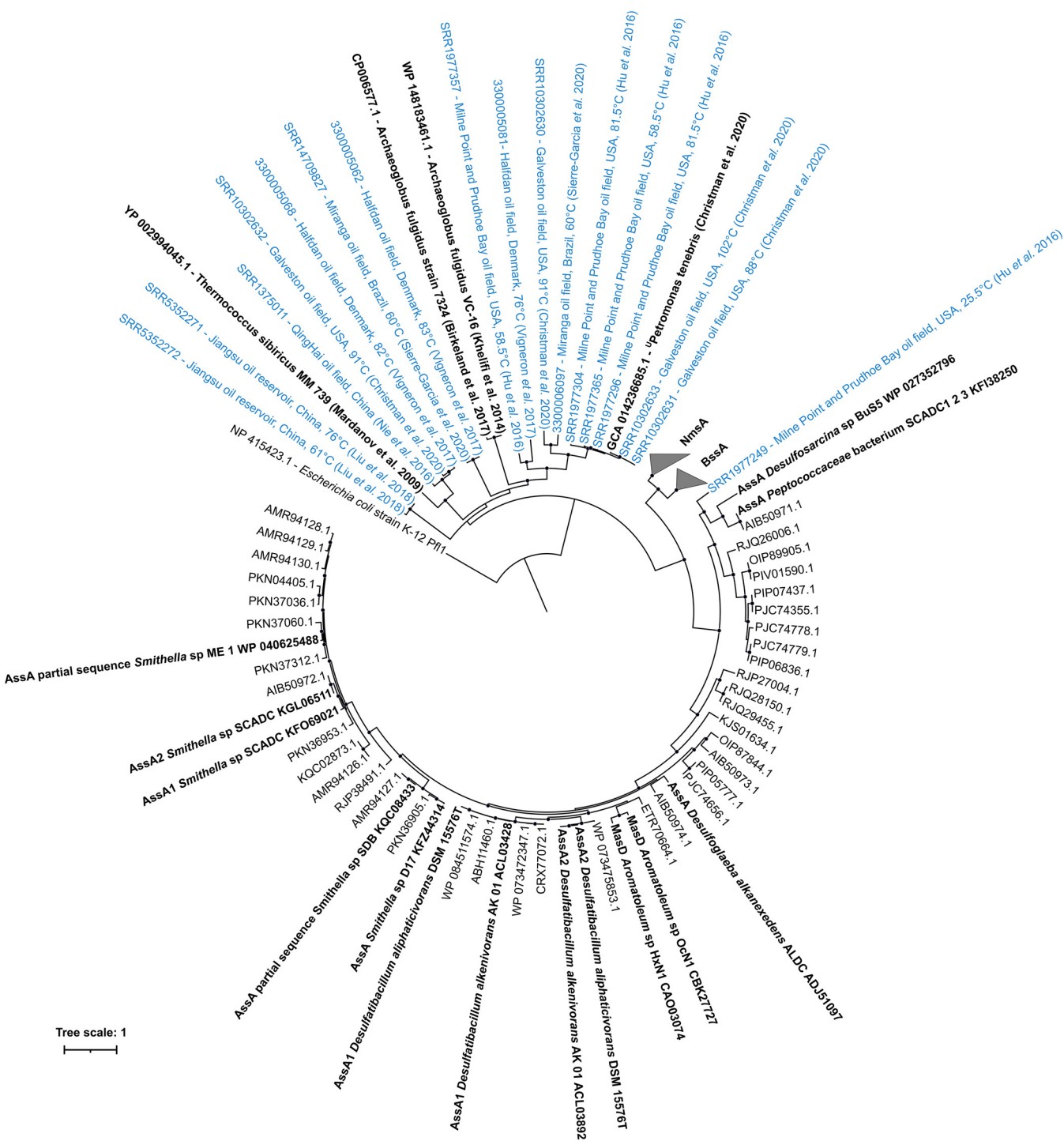

**FIG 5** Phylogeny of *assA*-like pyruvate formate-lyase (*pflD*) genes. Putative anaerobic alkane-activating pyruvate-formate lyase enzyme variants from 16 of the 25 metagenomes analyzed in this study (shown in blue) cluster together with homologous *pflD* gene sequences found in oil reservoir thermophiles [U]*Petromonas tenebris* (44), *Archaeoglobus fulgidus* strain 7324 (42, 43), and *Thermococcus sibiricus* strain MM 739 (45). *Archaeoglobus* and *Thermococcus* strains have been shown to degrade alkanes in pure culture (43, 45). These pflD gene sequences, as well as other experimentally verified alkane succinate synthase (AssA and MasD) genes, are shown in bold. Benzyl succinate synthase (BssA) and naphthyl-2-methyl-succinate synthase (NmsA) sequences are represented by collapsed clades. Black circles at the branch nodes indicate >80% bootstrap support (1,000 resamplings). Scale bar indicates a 10% sequence divergence as inferred from PhyML. The tree is rooted using Pfl from *E. coli*.

potential for anaerobic hydrocarbon biodegradation within the petroleum reservoir microbiome.

The detection of aerobic genes in many of the oil reservoir metagenomes could be considered surprising. Despite the possible influx of oxygen-bearing meteoric waters

(46, 47), anoxic conditions are understood to prevail in subsurface petroleum reservoirs (7). Interestingly, metagenomes containing genes for aerobic hydrocarbon activation were not restricted to samples associated with secondary oil recovery; long-chain alkane and monoaromatic dioxygenase enzymes are encoded in microbiomes from an offshore reservoir in the Gulf of Mexico (44) and an onshore reservoir in northeastern Brazil (48) that had not undergone secondary recovery before sampling for DNA analysis (Fig. 4). These results are consistent with observations of high proportions of genes for enzymes involved in aerobic hydrocarbon metabolism in various anaerobic hydrocarbon resource environments including oil reservoirs (29, 49) and may arise from the enrichment of initially minor groups of organisms capable of aerobic respiration coming into contact with air during sampling and sample transport. This is substantiated by genes encoding alkane hydroxylase (*alkB*) and monoaromatic dioxygenase (MAH) showing high-sequence identity (based on BLAST against the nonredundant nucleotide database) with *Marinobacter* spp., which are generally detected in low relative abundance *in situ* by 16S rRNA gene amplicon sequencing (Table S5). Enrichment of aerobic bacteria *ex situ* during sampling is ecologically similar to the proliferation of sulfate- or nitrate-reducing bacteria *in situ* in response to changing redox conditions, pointing more generally to an inactive or latent metabolic potential inherent to oil reservoir microbial communities.

**Provenance of the oil reservoir microbiome.** The extensive nature of this global data set of oil reservoir microbiomes suggests a pattern of subsurface biogeography featuring increasing community variance with greater geographic distance between reservoirs (Fig. 2). This raises questions about the provenance of microbes and the establishment of the global reservoir microbiome in a geologic context. Heuer et al. (50) postulated that vegetative cells and endospores that are deposited in surface sediments and undergo burial over geological timescales can then be revived under the right selective conditions. This is consistent with recent observations in a 1,180-m sediment core sampled during the International Ocean Discovery Program (IODP) Expedition 370 (51) and the concept of a microbial dispersal loop proposed for understanding the interplay of ecological principles of selection and migration in the subsurface (52, 53). Aside from geological processes such as sedimentation and petroleum fluid migration, natural dispersal vectors in the deep biosphere are limited (54). This evidence combines to suggest that microbial communities in oil reservoirs are inherited from populations that are present during the proximal deposition of sediments that eventually form reservoirs. Environmental selection during burial likely contributes to the differences observed here as a function of geography between oil reservoirs from different parts of the world.

In conclusion, DNA-based assessments of petroleum reservoirs continue to offer great potential for understanding difficult-to-access deep biosphere habitats and making important operational predictions in the face of changing environmental conditions. Designing and managing effective oil recovery strategies depend on the interplay between biogeochemical reactions and cycles catalyzed by microbiomes that are versatile and resilient to dramatic perturbations like seawater injection. Extensive screening of hundreds of samples by amplicon sequencing demonstrates the effects of environmental selection and biogeography in oil reservoirs globally. Gene-centric metagenomic analysis reveals that environmental selection is acting on a core functional biochemical potential based on identifying widespread genes for key processes such as sulfate reduction, sulfide oxidation, nitrate reduction, and methanogenesis. Less widespread detection of other genes, such as those suspected to catalyze anaerobic hydrocarbon biodegradation, raise important questions about these ecosystems and suggest that microbiome investigations will continue to deliver insights into microbial processes of industrial and ecological relevance.

## MATERIALS AND METHODS

**Data acquisition.** To determine the community composition and functional potential of the petroleum reservoir microbiome, 62 published studies with microbial assessments were selected based on the following criteria: (i) samples originating from a petroleum reservoir, (ii) sample collection points at the well head or directly associated infrastructure, and (iii) publicly available sequence data. Metadata including sample collection point and type, reservoir name, reservoir (and/or collection point) location, temperature and depth, and the use of primary or secondary (e.g., water or steam injection) production

were compiled directly from the respective publications (Table S1). In instances where geographic coordinates of the reservoir were not provided, latitude and longitude have been approximated based on the location of the reported oil field.

**16S rRNA gene sequence processing.** Raw high-throughput 16S rRNA gene amplicon sequence data were obtained from the National Center for Biotechnology Information's (NCBI) Sequence Read Archive (SRA; reference 55) by compiling sequence accession lists and implementing the *prefetch* and *fastq-dump* commands from the SRA Toolkit. The resulting amplicon data set comprises 295 sequence libraries from 201 different reservoir samples (i.e., 94 samples had amplicon libraries prepared using 2 different primer sets). In total, 53 million raw sequence reads and 62 Gb of sequence information were retained for analysis. Initial sequence processing was performed using the open-source VSEARCH version 2.11.1 (56). In instances where paired-end sequence data was available from the SRA, read pairs were merged based on a minimum read overlap length of 10 bp and a maximum permitted mismatch of 20% of the length of the overlap. Merged reads were filtered with a maximum expected error of 0.5 for all bases in the read, and minimum and maximum read lengths of 150 and 500 bp, respectively (informed by the reported amplicon size). Reads were dereplicated and annotated with their associated total abundance for each sample, before *de novo* chimera detection and removal using default parameters. Accounts of the reads retained at each processing stage are provided in Table S2. The quality-controlled sequences are available in the figshare online open access repository at https://doi.org/10.6084/m9.figshare.c.5801735.v2.

The studies compiled in this meta-analysis were published between 1993 and 2020, representing a period during which sequencing technology and accepted standards evolved. Reliable assessments of microbiomes are dependent on high-quality data sets with accurate sequence information. Despite applying quality control steps designed to include a larger range of studies and enable a meta-analysis with maximum breadth, in some instances as few as 1.7% of the raw reads in a given library and 8.4% of raw reads in an entire study could be retained. This highlights how data quality can be an important and sometimes unresolved issue in microbiology investigations. It also demonstrates how different studies have applied different quality standards to DNA sequencing results. The widely accepted and used quality control parameters in this meta-analysis ensures robust conclusions are drawn from the analysis.

Quality-filtered 16S rRNA gene amplicon sequences from each high-throughput library were concatenated into a single file for further processing. To account for the different target hypervariable regions chosen within studies and between studies, sequences with consistent taxonomic classification were considered members of the same phylotype, allowing all libraries to be compared regardless of PCR methodology. Taxonomy was assigned using Mothur version 1.41.3 (57) and Ribosomal Database Project's k-mer-based naive Bayesian classifier (58) with the SILVA SSU Ref NR version 138 database (59). A bootstrap cutoff value was set to 80% to return only taxonomies above this confidence threshold. A custom R script written in base R version 3.6.1 (60) parsed the output taxonomic assignments into a sample-by-phylotype table that accounted for earlier dereplication of the sequences. Sequence processing included the removal of singleton phylotypes, phylotypes with no domain-level taxonomic classification (likely artifacts of sequence preprocessing), and phylotypes classified as *Vertebrata*, *Mitochondria*, and *Chloroplast*. High-throughput sequencing libraries with <1,000 reads after quality filtering were removed from the data set. To limit the loss of data, subsampling was only used for alpha-diversity calculations and for comparisons of the effect of variable library sizes (results are reported alongside non-subsampled libraries in specific instances as described herein). Subsampling for alpha diversity was performed without replacement to 1,000 reads using the *phyloseq* R package (61). For all other analyses, subsampling was not performed to maximize the depth of information extracted from amplicon sequencing libraries.

Oil reservoir amplicon sequencing studies have also employed clone libraries of amplified PCR products. A total of 2,850 amplicon sequencing reads derived from 48 different studies were downloaded from NCBI's GenBank database (62) and analyzed separately from the high-throughput 16S rRNA gene amplicon sequence data described above to account for the various sampling depth and data outputs. No sequence processing steps were implemented prior to the analysis of these "low-throughput" amplicon libraries due to error checking performed on submission to GenBank; however, a reverse complement sequence was used for taxonomy in instances where this produced a better classification. Analyzed clone sequences are compiled in the figshare online open-access repository at https://doi.org/10.6084/m9.figshare.c.5801735.v2. Taxonomy was assigned using Mothur version 1.41.3 (57) and Ribosomal Database Project's k-mer-based naive Bayesian classifier (58) with the SILVA SSU Ref NR version 138 database (59). A bootstrap cutoff value was set to 80%. The use of the same taxonomic classification tool, database, and parameters meant phylotype-level community structures between high-throughput and low-throughput amplicon libraries were comparable. Counts of the respective classifications were used to assess prevalence across the data set at various taxonomic resolutions.

**Metagenome processing.** The metagenomic data set used here comprised sequences from 25 metagenome libraries from nine different reservoirs. To assemble this data set, 16 libraries accounting for 457 million raw sequence reads were downloaded from NCBI's SRA (55). An additional 9.6 million assembled reads from nine libraries were downloaded from the Integrated Microbial Genomes & Microbiomes system (IMG/M; reference 63) when raw sequence data were not publicly available (Table S2). In total, 448 Gb of metagenomic sequence information was compiled. Raw, unassembled reads were quality controlled by trimming technical sequences (primers and adapters) and low-quality additional bases and filtering artifacts (phiX), low-quality reads, and contaminated reads using BBDuk (BBTools suite, http://jgi.doe.gov/data-and-tools/bbtools). Ribosomal rRNA genes in quality-controlled reads were reconstructed and classified using phyloFlash (64) with mapping against the SILVA SSU Ref NR version 138 database (59). Ribosomal rRNA genes in assembled contigs were identified using rRNAFinder (https://github.com/xiaoli

-dong/rRNAFinder) after contigs smaller than 500 bp were removed, and taxonomy was assigned using Mothur version 1.41.3 (57) with the SILVA SSU Ref NR version 138 database (59). Detailed accounts of the reads retained during processing and assembly statistics are provided in Table S2. Known challenges associated with 16S rRNA gene assembly from metagenomes (65) likely account for the lower gene counts in prior assembled data compared to unassembled data. For the quality-controlled reads not already assembled, the metagenome libraries were assembled using MEGAHIT version 1.2.2 (66) with default parameters, and contigs <500 bp were removed.

Gene-centric metagenomic analysis assessed the functional potential of the community as a whole. By focusing on genes rather than assembling whole provisional genomes much larger proportions of the metagenomic libraries are retained for analysis. Protein coding genes in each assembly were predicted using Prodigal version 2.6.3 (67). Predicted gene sequences were compiled and made available in the figshare online open access repository at https://doi.org/10.6084/m9.figshare.c.5801735.v2. Predicted genes were annotated with KO using GhostKOALA (68). Metabolic pathways from KO assignments were reconstructed using KEGG decoder (69). The KEGG Level C pathway was assigned by matching to KEGG BRITE database hierarchy (file ko00001.keg). Sporulation genes were identified by manual searching for KO numbers of specific marker genes. Genes involved in aerobic and anaerobic activation of hydrocarbon compounds (i.e., indicators of hydrocarbon biodegradation capability) were annotated using the CANT-HYD database of phylogeny-derived hidden Markov models (27). The CANT-HYD trusted cutoff domain score was used to annotate hydrocarbon activation functions reported in Table S7 and Fig. 4. For the identification of *assA*-like pyruvate formate lyase (*pflD*) genes, no domain-level score cutoff was used, and the highest scoring sequence for each of the respective metagenomes was retained for phylogenetic analysis (Fig. 5). Sequences sharing homology with *assA*-like pyruvate formate lyases (*pflD*) in metagenomes, together with experimentally verified and hypothetical *assA*, *bssA*, and *nmsA* genes, were aligned using Clustal Omega (70) before inference of a maximum likelihood phylogenetic tree using FastTree (71). Phylogenetic trees were annotated using iTOL version 5.5 (72).

**Statistical analysis and data visualization.** Statistical analyses and visualization were performed using base R version 3.6.1 (60) or the specific R packages as indicated below. Richness and alpha-diversity metrics were calculated for subsampled high-throughput 16S rRNA gene sequence libraries using *phyloseq* (61). Spearman's rank correlation coefficient assessed correlations between alpha-diversity indices and two-group variables. Kruskal-Wallis tests assessed alpha-diversity variance with respect to multigroup and nonnumeric variables.

Before beta-diversity calculations the nonsubsampled and subsampled high-throughput 16S rRNA gene amplicon data sets were split according to the implied and probable use of universal or archaeal primers; libraries comprising ≥75% archaea (based on relative sequence abundance) are defined here as archaeal libraries and the remainder defined as universal libraries indicating nontaxon-specific 16S rRNA gene amplification. Nonmetric multidimensional scaling (NMDS) of Bray-Curtis dissimilarity between high-throughput 16S rRNA gene-based communities was calculated from relative abundance data using *phyloseq* (61) and visualized using *ggplot2* (73). Statistical differences between sample community dissimilarities (Bray-Curtis) in relation to nonnumeric variables were assessed using permutational multivariate analysis of variance (PERMANOVA) tests in *vegan* (74). Biogeographical patterns were assessed using Mantel tests of sample community dissimilarities (Bray-Curtis) and the Euclidean distance between environmental parameters or haversine (geographic) distance between locations. Haversine distance was calculated using the *geosphere* package (75), and Mantel tests were performed using *vegan* (74).

Microbial indicator species analysis, designed to test the association of a single taxon with an environment through multilevel pattern analysis, was used to identify phylotypes that best represent a specific environmental condition based on both phylotype presence/absence and relative abundance patterns. Indicator phylotypes were calculated using the *multipatt* function of the *indicspecies* package in R, employing a point-biserial correlation index (21). Tests were performed on nonsubsampled libraries (see Table S1 for temperature subsets).

## SUPPLEMENTAL MATERIAL

Supplemental material is available online only.

**TABLE S1**, XLSX file, 0.1 MB.
**TABLE S2**, XLSX file, 0.1 MB.
**TABLE S3**, XLSX file, 0.1 MB.
**TABLE S4**, XLSX file, 0.02 MB.
**TABLE S5**, XLSX file, 6.8 MB.
**TABLE S6**, XLSX file, 0.1 MB.
**TABLE S7**, XLSX file, 0.9 MB.

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
