## [Reviewer comments · mSystems]

Environmental selection and biogeography shape the microbiome of subsurface petroleum reservoirs

Daniel Gittins, Srijak Bhatnagar, and Casey Hubert

Corresponding Author(s): Daniel Gittins, University of Calgary

Review Timeline:

Submission Date:	September 8, 2022
Editorial Decision:	October 21, 2022
Revision Received:	December 8, 2022
Accepted:	January 15, 2023

Editor: Jacqueline Goordial

Reviewer(s): The reviewers have opted to remain anonymous.

Transaction Report:

DOI: <https://doi.org/10.1128/msystems.00884-22>

October 21, 2022

Dr. Daniel A. Gittins
University of Calgary
Biological Sciences
2500 University Drive NW
Calgary, Alberta T2N1N4
Canada

Re: mSystems00884-22 (Environmental selection influences the microbiome of subsurface petroleum reservoirs)

Dear Dr. Daniel A. Gittins:

Thank you for submitting your manuscript to mSystems. We have completed our review and I am pleased to inform you that, in principle, we expect to accept it for publication in mSystems. However, acceptance will not be final until you have adequately addressed the reviewer comments.

I have reviewed the text as well, and agree with both reviewers who indicated that the manuscript could be significantly improved by editing the existing text for length, repetition and clarity. In some instances the reviewers have indicated the potential to include additional analysis (eg. outgroup inclusion). In a future response to reviewers document, please identify if these analyses have been included, and if not, please address the reviewers concerns about defining a 'core' microbiome with out the inclusion of an outgroup.

Preparing Revision Guidelines

Sincerely,

Jacqueline Goordial

Editor, mSystems

Journals Department
Reviewer comments:

Reviewer #1 (Comments for the Author):

General

In this study, the authors perform a meta-analysis on public sequencing data (amplicon and shotgun metagenomic) from oil reservoirs. They suggest that these provide evidence that there are not core taxa shared across reservoirs, but there are core "functions".

Overall, the goals and importance are not well defined in this work. I don't really understand why it's been done or what this meta-analysis contributes that the original studies did not. I think there are potentially interesting insights to pull out of these data, but as presented, I was not convinced of the importance of this work. No clear hypothesis/objective is stated in the introduction. The discussion is very list-like and unfocused.

I also feel that a lot of the conclusions come across as very wishy washy, because there is no point of reference. For instance, to say that functions are "conserved" within the set of samples you looked at, you would need to demonstrate that this is somehow different from a set of outgroup samples. Methodological choices were similarly wishy washy, as the logic behind them was often not explained.

While I feel it is possible for the authors to rework this, it would be a substantial effort that (to me) is beyond a simple revision.

Specific

Title: "Environmental selection" is pretty vague. I would come up with something that is closer to what you are actually showing here.

Line 11: Extreme in what way?

Line 12-14: The fact that systematic studies of core taxa are limited does not mean that such studies are important. Need to make the case for the importance of this work and the gap it fills. Nobody has conducted systematic studies on how many bricks make up the houses on my block, but I also don't think anybody needs to do that. Convince us of the importance.

Line 16-18: Importance of any of this?

Line 18-20: Functional core at what hierarchical level? Needs more specificity to help us understand. For instance, all bacteria living in the ocean and in soil engage in protein synthesis, but I wouldn't consider that to be a surprising or important finding. Help us to see what is new/interesting here.

Line 20-22: And? Meaning?

Line 22: What do you mean by metabolic redundancy? Redundant compared to what? If there are two organisms capable of the same function, is that redundant? Is that important?

Line 23-25: From this abstract, I don't see how.

Line 28-30: Are they more similar than they are to some relevant outgroups? How do you define "consistent"? Relative to what?

Line 61-72: This is not a very good sell on the importance of a core microbiome. In many scenarios, it may specifically be the non-core microbes that are important. For instance, if we grow corn across a wide geographic range, is it the consistent microbes (those that are ubiquitous) that matter? Are the ones that are more locally adapted/specific? Hard to know, but it's not a given that the "core" is most important...I certainly don't agree with the last sentence here, or at least, a specific case is not made for why this should be true.

Line 80-84: No specific hypotheses or objectives defined. Not clear what gap this work is filling or why it should be done. Not

made clear what the meta-analysis adds on what has already been shown.

Line 117: How would your data have changed if you used a different quality cutoff? Why to you was it more important to have higher quality per-read data than to include more sequences per sample? This choice is not obvious so needs to be justified. I would say that retaining only 1.7% of sequences from a given sample could be really problematic, depending on their sequencing depth.

Line 121-124: If you're already doing a phylotype analysis, it's not clear to me why you would need to use stringent sequence quality cutoffs...why not relax those for studies using older technologies etc.? Needs to be justified.

Line 144-147: Again, this difference in treatment suggests maybe it would have been useful to relax quality filtering for certain studies.

Line 151-153: How so? Explain.

Line 156: Is geographic dispersion similar in these samples compared to amplicon samples? Not described.

Line 234: Do you mean 4-8%?

Line 263: This section is actually more interesting than the discussions about "core" and what is the same. Understanding what changes with different key environmental factors is potentially an important gap to fill (if not already known).

Line 313: How do you define "highly conserved"? Did you use any outgroups?

Line 388: Was this your hypothesis? Stated where?

Line 393: You don't know that they were not universal...could be below detection and you mentioned having very few reads pass quality filtering in some cases.

Line 399: Defined how? All bacteria possess 16S rRNA genes...is that considered a functional core?

Line 402-404: Strongly disagree with this statement. How is this statement different from what you could say about microbiomes in any environment?

Figure 2BD: How are we to interpret 0 km? This is referring to replicate samples from the same location?

Figure 3: No legend, have no idea how to interpret.

Reviewer #3 (Comments for the Author):

I really enjoyed reading the paper of Gittins and co-authors about the microbiome of subsurface petroleum reservoirs. The authors compiled 343 16S rRNA data sets and 25 metagenomic libraries to identify its core microbial taxa and associated genomic attributes. I very much liked how the different data sets were merged. The detailed description about processing sequence data information was also very helpful.

Surprisingly, a core microbiome could not be identified. Even genes for anaerobic hydrocarbon degradation could not be considered as core biogeochemical functions. Instead, depth and temperature seem to drive these specific subsurface communities. The gene-centric metagenomics analyses revealed functional core featuring carbon acquisition and energy conservation strategies. Although these findings are not exceptional in their novelty, this is one of the first comprehensive meta-analysis on subsurface core microbiomes which may guide engineering interventions and warrants priority publication.

In general, the manuscript is very well written. Just some parts in the results and discussion were a bit lengthy and contained repetitions that distract from the main findings. I would also prefer a clearer separation between results and discussion (see my comments below).

Unusual for me, I don't have many comments for the authors to consider.

1) Lines 73-84: Please shorten this text. Provide a clear aim for this meta-analysis.

2) Line 90: It would be very helpful, if you can provide the number and some more details of the samples you used for the meta-analysis. Most readers will not dive into Table S1 to find this information.

3) Line 114: Please mention the years of the publications.

4) Line 245: What means N.B.?

5) Line 263: How do come to the number of 368? It is not the sum of 295 and 48? Please add more information here.

6) Lines 322-337: There is no need to include references here and to discuss these detailed findings. Acetogenesis is picked up later in the discussion with a very similar wording.

7) Line 344: Please delete: "if sulfate is present". This is not necessary.

- 8) Line 377: Please remove the references and shorten this part.
- 9) Line 419. Please delete the following sentence. It is too trivial.
- 10) Line 430: Which electron acceptors? Please be more specific here. In general, this paragraph needs some input It contains too many very general statements. You could also just delete these sentences, as the topic is picked up again in the next section.
- 11) Line 453: Just delete this sentences. You just repeat results here.
- 12) Line 469: Delete the first part of the sentence.
- 13) Line 476: What are anaerobic genes? Please rephrase.
- 14) Provenance of the oil reservoir microbiome: Please shorten the whole section, avoid redundancies, do not repeat methods, etc. Just focus on your findings.
- 15) Line 511: Dramatically? Please rephrase.
- 16) Please shorten the conclusion section.

Reviewer #1 (Comments for the Author):

General

In this study, the authors perform a meta-analysis on public sequencing data (amplicon and shotgun metagenomic) from oil reservoirs. They suggest that these provide evidence that there are not core taxa shared across reservoirs, but there are core "functions".

We appreciate this Reviewer's time and attention, and believe these suggestions have resulted in considerable improvements to the manuscript.

Overall, the goals and importance are not well defined in this work. I don't really understand why it's been done or what this meta-analysis contributes that the original studies did not. I think there are potentially interesting insights to pull out of these data, but as presented, I was not convinced of the importance of this work. No clear hypothesis/objective is stated in the introduction. The discussion is very list-like and unfocused.

Thank you for this feedback. The introduction now states a clear hypothesis that petroleum reservoirs harbour a core microbiome at lines 103-105. This allows the results of the manuscript to falsify the hypothesis with respect to the absence of a taxonomic core.

We appreciate this suggestion, which gives the manuscript a better overarching structure.

The manuscript now more clearly highlights the importance of performing a meta-analysis. Asking and answering questions about core microbiomes – as our manuscript does – cannot be achieved by considering the 62 original studies in isolation.

We value the feedback about the Discussion, which we have modified and shortened to improve readability.

I also feel that a lot of the conclusions come across as very wishy washy, because there is no point of reference. For instance, to say that functions are "conserved" within the set of samples you looked at, you would need to demonstrate that this is somehow different from a set of outgroup samples. Methodological choices were similarly wishy washy, as the logic behind them was often not explained.

The manuscript now more clearly anchors the study against a commonly accepted definition of core microbiomes in different environments, as outlined by the Shade & Handlesman (2011) paper that has been cited over 800 times. We agree that the term "conserved" can cause confusion about our findings and their implications, and never intended to imply that these functions are unique to this ecosystem. To avoid this confusion, we have updated the manuscript to instead define these as "shared functions". This should clarify why an outgroup is not necessary.

While I feel it is possible for the authors to rework this, it would be a substantial effort that (to me) is beyond a simple revision.

Specific

Title: "Environmental selection" is pretty vague. I would come up with something that is closer to what you are actually showing here.

We have changed the title to:

"Environmental selection and biogeography shape the microbiome of subsurface petroleum reservoirs"

Adding "biogeography" provides a better description of the manuscript's novel contribution.

Line 11: Extreme in what way?

We agree that this term should have been qualified. We have added a sentence (lines 55-57) to the Introduction to support our statement that petroleum reservoirs are extreme environments.

Line 12-14: The fact that systematic studies of core taxa are limited does not mean that such studies are important. Need to make the case for the importance of this work and the gap it fills. Nobody has conducted systematic studies on how many bricks make up the houses on my block, but I also don't think anybody needs to do that. Convince us of the importance.

We now add more emphasis to variation in microbial community composition across a range of environmental conditions (line 16). This still allows for an assessment of the presence (or in this case absence) of a core microbiome, but adds context to why systematic studies of this nature are important from an ecological and industrial perspective.

Line 16-18: Importance of any of this?

We have provided an expanded statement that explains the importance of variation among reservoirs with different physicochemical characteristics, and with geographic distance as *"highlighting environmental selection and biogeography in these deep biosphere habitats"* (lines 22-23).

Line 18-20: Functional core at what hierarchical level? Needs more specificity to help us understand. For instance, all bacteria living in the ocean and in soil engage in protein synthesis, but I wouldn't consider that to be a surprising or important finding. Help us to see what is new/interesting here.

This part has been re-written (lines 24-26) to emphasize that the functions of interest pertain to biogeochemical cycling in the deep biosphere in general and oil reservoirs in particular.

Line 20-22: And? Meaning?

We add meaning to the finding that genes for anaerobic hydrocarbon degradation are observed only in a subset of the samples by now noting that hydrocarbons represent an abundant carbon source in oil reservoir environments, supporting why we think this observation is unexpected and worth highlighting (lines 26-29).

Line 22: What do you mean by metabolic redundancy? Redundant compared to what? If there are two organisms capable of the same function, is that redundant? Is that important?

Thank you for this feedback. Upon re-evaluation we agree that we have not shown metabolic redundancy (different phylotypes performing similar functional role in the community). Removing this has improved the manuscript. We now provide a new ending that highlights the factors controlling subsurface microbial community assembly (lines 29-31).

Line 23-25: From this abstract, I don't see how.

We agree that we have not shown obvious relevance to engineering. Thank you for noting this. We now provide a new ending, as mentioned in the comment above, highlighting controls on subsurface microbial community assembly.

Line 28-30: Are they more similar than they are to some relevant outgroups? How do you define "consistent"? Relative to what?

An outgroup is not required to qualify a statement that taxonomy varies between reservoirs or that similar functions are seen across a range of environmental conditions. It is important to note that a core microbiome does not imply distinct microbes that are found nowhere else, but rather defines shared taxa and functions across samples with different parameters. Therefore, these taxa or functions are not described as unique to this ecosystem, but rather as being shared across geographically distinct reservoir samples.

An outgroup would be useful if the aim is to define taxa or functions unique to this environment. We also think that it is possible for any outgroup sample (e.g., from a surface environment) to contain microbes (taxa/functions) derived from oil reservoirs, owing to dispersal processes that expel the oil reservoir microbiome – a topic we have published on recently (Gittins *et al.* 2022 *Science Advances*, 8, eabn3485). For these reasons, inclusion of an outgroup may add confusion.

As mentioned previously, we have replace “conserved” with “shared” to circumvent the important issues raised here.

Line 61-72: This is not a very good sell on the importance of a core microbiome. In many scenarios, it may specifically be the non-core microbes that are important. For instance, if we grow corn across a wide geographic range, is it the consistent microbes (those that are ubiquitous) that matter? Are the ones that are more locally adapted/specific? Hard to know, but it's not a given that the "core" is most important...I certainly don't agree with the last sentence here, or at least, a specific case is not made for why this should be true.

We have added and explanation (line 79) with references (line 84) to support our statement that shared taxa often mediate ecologically and functionally important processes in environments.

Lines 84-87 (i.e., next sentence) supports that absence of a “core” can result in shifts in functioning, which acknowledges your point that the non-core microbes are also important.

New references:

Lundberg DS, Lebeis SL, Paredes SH, Yourstone S, Gehring J, Malfatti S, et al. (2012) Defining the core *Arabidopsis thaliana* root microbiome. *Nature* 488: 86–90.

Ainsworth TD, Krause L, Bridge T, Torda G, Raina JB, Zakrzewski M, et al. (2015) The coral core microbiome identifies rare bacterial taxa as ubiquitous endosymbionts. *ISME J* 10: 2261–2274.

Line 80-84: No specific hypotheses or objectives defined. Not clear what gap this work is filling or why it should be done. Not made clear what the meta-analysis adds on what has already been shown.

Thank you for this comment. We have added text (lines 103-105) to clearly state the aims of the meta-analysis (i.e., to "*test the hypothesis that petroleum reservoirs contain a core microbiome with respect to both taxonomy and biogeochemical functions*"). This edit now provides more focus to our manuscript.

Line 117: How would your data have changed if you used a different quality cutoff? Why to you was it more important to have higher quality per-read data than to include more sequences per sample? This choice is not obvious so needs to be justified. I would say that retaining only 1.7% of sequences from a given sample could be really problematic, depending on their sequencing depth.

We specified our QC steps and based upon the widely accepted and used standards of 16S rRNA gene amplicon sequencing data analyses. These are not considered extremely stringent, but still ensure a high quality of data are included for the analysis. The observation articulated in the methods about some studies having low data retention reflects more on the quality of data being published and deposited in public databanks. Accounts of the reads retained from each study are provided in Table S2, as indicated at line 137.

Line 121-124: If you're already doing a phylotype analysis, it's not clear to me why you would need to use stringent sequence quality cutoffs...why not relax those for studies using older technologies etc.? Needs to be justified.

At line 143 we indicate that sequence quality control steps were designed to include a larger range of studies. Relaxing these beyond our cutoffs would likely introduce spurious sequences, affecting taxonomic classification and therefore the reliability of our phylotype analysis.

This is also an important issue for other users of this dataset (including ourselves) once it is published and publicly available in this compiled form. We see this study as being a useful resource for others with other questions about oil reservoir microbiomes, and wish to ensure the integrity of the dataset by sticking with these cutoffs.

Line 144-147: Again, this difference in treatment suggests maybe it would have been useful to relax quality filtering for certain studies.

Sequences submitted to GenBank using the wizard are automatically processed and checked for chimeras, vector contamination, low quality sequence, and other problems (Sayers *et al.* 2019; <https://www.ncbi.nlm.nih.gov/pmc/articles/PMC6323954/>). On this basis, quality filtering is

not required for clone sequences in the same way as for high throughput sequences submitted to the SRA. We have added an explanation of this at line 176.

Line 151-153: How so? Explain.

Sentence edited (line 183) to indicate that taxonomic classification used the same tool, database, and parameters for consistency between high and low throughput amplicon library analyses.

Line 156: Is geographic dispersion similar in these samples compared to amplicon samples? Not described.

We have added text to describe that the metagenomes are derived from samples across nine different reservoirs (line 188)

Line 234: Do you mean 4-8%?

Thank you. Corrected (line 268)

Line 263: This section is actually more interesting than the discussions about "core" and what is the same. Understanding what changes with different key environmental factors is potentially an important gap to fill (if not already known).

Thank you for noting this. We have adjusted the manuscript abstract to better highlight how an absence of a core microbiome is likely related to variation across different environmental conditions.

Line 313: How do you define "highly conserved"? Did you use any outgroups?

An outgroup is not required to qualify the statement that there is genetic similarity across the global petroleum reservoir microbiome. However, we agree that "highly" is a relative term. To avoid this being a contentious statement, we have removed this from the sentence (line 363).

Line 388: Was this your hypothesis? Stated where?

As per other suggestions, we have added a hypothesis in the Introduction (lines 103-105) as well as refocused the manuscript to emphasize variability across environmental gradients. We have added additional references to support the regularly used definition of a core microbiome (lines 437-438).

Line 393: You don't know that they were not universal...could be below detection and you mentioned having very few reads pass quality filtering in some cases.

The statement is true as it uses the phrase "indicated" to describe what the data shows, i.e., no universally present phyla in metagenomic libraries. Furthermore, this sentence refers to metagenomic libraries, where there was no substantial data loss.

Line 399: Defined how? All bacteria possess 16S rRNA genes...is that considered a functional core?

We have reworded this sentence to avoid the tricky connotations of referring to a 'core'. Instead,

we claim "*shared biochemical functions*" (line 449).

Line 402-404: Strongly disagree with this statement. How is this statement different from what you could say about microbiomes in any environment?

We agree. The statement has been modified to "*Diverse genes encode functions for processes such as fermentation, sulfate reduction, hydrocarbon biodegradation and methanogenesis in the oil reservoir microbiome*" (lines 452-455).

Figure 2BD: How are we to interpret 0 km? This is referring to replicate samples from the same location?

That is correct. This is now described in the figure caption: "*A geographic distance of '0' indicates reservoirs at the same location.*"

Figure 3: No legend, have no idea how to interpret.

The figure shows an overview of variability between taxonomic and functional groups. A legend is not required to show this. However, to ensure the reader has all available information, we have added a new sheet to Table S7 that provides the raw data for the KEGG Level C pathway assignments.

Reviewer #3 (Comments for the Author):

I really enjoyed reading the paper of Gittins and co-authors about the microbiome of subsurface petroleum reservoirs. The authors compiled 343 16S rRNA data sets and 25 metagenomic libraries to identify its core microbial taxa and associated genomic attributes. I very much liked how the different data sets were merged. The detailed description about processing sequence data information was also very helpful.

We appreciate this constructive feedback, thank you.

Surprisingly, a core microbiome could not be identified. Even genes for anaerobic hydrocarbon degradation could not be considered as core biogeochemical functions. Instead, depth and temperature seem to drive these specific subsurface communities. The gene-centric metagenomics analyses revealed functional core featuring carbon acquisition and energy conservation strategies. Although these findings are not exceptional in their novelty, this is one of the first comprehensive meta-analysis on subsurface core microbiomes which may guide engineering interventions and warrants priority publication.

We agree that the lack of a taxonomic core is interesting and surprising. In the revised version we have attempted to underscore this aspect of the study by framing and then falsifying the hypothesis of a core microbiome.

We appreciate this reviewer's endorsement and suggestion of prioritising the manuscript for publication.

In general, the manuscript is very well written. Just some parts in the results and discussion were a bit lengthy and contained repetitions that distract from the main findings. I would also prefer a clearer separation between results and discussion (see my comments below). Unusual for me, I don't have many comments for the authors to consider.

We have attempted to shorten and streamline the manuscript, as articulated in the responses provided here.

1) Lines 73-84: Please shorten this text. Provide a clear aim for this meta-analysis.

We have edited the final paragraph of the introduction to be more succinct, and to clearly state that we used this dataset to "*test the hypothesis that petroleum reservoirs contain a core microbiome with respect to both taxonomy and biogeochemical functions.*" (Lines 103-105).

2) Line 90: It would be very helpful, if you can provide the number and some more details of the samples you used for the meta-analysis. Most readers will not dive into Table S1 to find this information.

Thank you for noting this. We have provided revisions throughout to provide clear details of the number of samples and libraries included in this study. The total number of published studies (62) included is now mentioned at line 114. Other relevant details are included at intervals throughout the methods (e.g., lines 126-128, 171, 188) in the results section (e.g., line 263, 333-335).

3) Line 114: Please mention the years of the publications.

The range of publication years are now provided (line 140).

4) Line 245: What means N.B.?

To avoid confusion we have deleted N.B. (i.e., nota bene or "note well"), which is not required for the sentence to hold its meaning.

5) Line 263: How do come to the number of 368? It is not the sum of 295 and 48? Please add more information here.

We have addressed this issue throughout by making clear statements about the number of samples and libraries which we are referring to. Thank you for bringing this to our attention. This comment has led to a significantly improved manuscript. Given that the number reported at this line is misleading, we have opted to remove it from the manuscript.

6) Lines 322-337: There is no need to include references here and to discuss these detailed findings. Acetogenesis is picked up later in the discussion with a very similar wording.

We appreciate this suggestion, and have removed this to avoid unnecessary repetition.

7) Line 344: Please delete: "if sulfate is present". This is not necessary.

Removed.

8) Line 377: Please remove the references and shorten this part.

We have removed the accessory information about hydrocarbon metabolism in reservoirs worldwide over geological timescales as well as the supporting references (lines 425-427).

9) Line 419. Please delete the following sentence. It is too trivial.

Removed.

10) Line 430: Which electron acceptors? Please be more specific here. In general, this paragraph needs some input. It contains too many very general statements. You could also just delete these sentences, as the topic is picked up again in the next section.

To avoid repetition, the sentence pertaining to the availability of electron acceptors was removed (lines 481-484). We agree that this was unnecessary content.

11) Line 453: Just delete this sentence. You just repeat results here.

Sentence removed and slight modification offered to improve the flow of the opening sentence (line 505).

12) Line 469: Delete the first part of the sentence.

Removed (line 522)

13) Line 476: What are anaerobic genes? Please rephrase.

Corrected to "*genes for aerobic hydrocarbon activation*" (line 529).

14) Provenance of the oil reservoir microbiome: Please shorten the whole section, avoid redundancies, do not repeat methods, etc. Just focus on your findings.

Thank you for this comment. We have removed the second paragraph of this section within the Discussion to ensure that our examination of provenance remains focussed. As per this reviewer's more general suggestions around length, we think this contributes to a cleaner and more concise message highlighting the most interesting points about assembly of the deep biosphere.

15) Line 511: Dramatically? Please rephrase.

No longer included in revised Discussion section, as per the point above.

16) Please shorten the conclusion section.

Content removed, e.g., lines 593-595, to provide a more succinct conclusion.

January 15, 2023

Dr. Daniel A. Gittins
University of Calgary
Biological Sciences
2500 University Drive NW
Calgary, Alberta T2N1N4
Canada

Re: mSystems00884-22R1 (Environmental selection and biogeography shape the microbiome of subsurface petroleum reservoirs)

Dear Dr. Daniel A. Gittins:

Your manuscript has been accepted, and I am forwarding it to the ASM Journals Department for publication. For your reference, ASM Journals' address is given below. Before it can be scheduled for publication, your manuscript will be checked by the mSystems production staff to make sure that all elements meet the technical requirements for publication. They will contact you if anything needs to be revised before copyediting and production can begin. Otherwise, you will be notified when your proofs are ready to be viewed.

If you would like to submit a potential Featured Image, please email a file and a short legend to mssystems@asmusa.org. Please note that we can only consider images that (i) the authors created or own and (ii) have not been previously published. By submitting, you agree that the image can be used under the same terms as the published article. File requirements: square dimensions (4" x 4"), 300 dpi resolution, RGB colorspace, TIF file format.

We recognize that the video files can become quite large, and so to avoid quality loss ASM suggests sending the video file via <https://www.wetransfer.com/>. When you have a final version of the video and the still ready to share, please send it to mSystems staff at mssystems@asmusa.org.

Sincerely,

Jacqueline Goordial
Editor, mSystems

Journals Department
E-mail: mSystems@asmusa.org